# Pricing Strategies of Logistics Distribution Services for Perishable Commodities

**Tao Li [1,2], Yan Chen [1] and Taoying Li [1,*]** 

[1]   School of Maritime Economics and Management, Dalian Maritime University, Dalian 116026, China;
      litao@wti.ac.cn (T.L.); chenyan@dlmu.edu.cn (Y.C.)
[2]   China Waterborne Transport Research Institute, Beijing 100088, China
*    Correspondence: ytaoli@126.com; Tel.: +86-155-6680-2152

**Abstract:** The problem of pricing distribution services is challenging due to the loss in value of product during its distribution process. Four logistics service pricing strategies are constructed in this study, including fixed pricing model, fixed pricing model with time constraints, dynamic pricing model, and dynamic pricing model with time constraints in combination with factors, such as the distribution time, customer satisfaction, optimal pricing, etc. By analyzing the relationship between optimal pricing and key parameters (such as the value of the decay index, the satisfaction of consumers, dispatch time, and the storage cost of the commodity), it is found that the larger the value of the attenuation coefficient, the easier the perishable goods become spoilage, which leads to lower distribution prices and impacts consumer satisfaction. Moreover, the analysis of the average profit of the logistics service providers in these four pricing models shows that the average profit in the dynamic pricing model with time constraints is better. Finally, a numerical experiment is given to support the findings.

**Keywords:** perishable goods; value loss; pricing strategy; logistics distribution

## 1. Introduction

Perishable goods are goods that are easily rotting, decomposing, and damaged during the process of logistics distribution. Such goods are generally characterized by uncertain demand, low residual inventory value, short sales cycle, and so on [1]. Supply chain for perishable goods, including the movement and storage of raw materials, work-in-process inventory, and finished goods from point of origin to point of consumption, is obviously very complex. We only focus on the issue of pricing logistics distribution services for perishable goods, and other issues will not be covered in this paper. The logistics distribution of perishable goods is clearly more complex than that of traditional goods because of uncertainty in all aspects of demand and distribution [2–6], the reasons can be interpreted as two points. On the one hand, traditional goods do not have special requirements for delivery time and storage environment. However, there is a higher time constraint requirement for perishable goods because goods tend to deteriorate or storage requires special requirements [2], which will increase the cost of distribution cost. On the other hand, customers are willing to pay higher prices for fast delivery perishable goods, therefore, unified price conflicts with customer demands and dynamic personalized price is most effective owing to different delivery time requirement [3–6], and the cost of delayed delivery also simultaneously needs to be considered [2]. During the process of logistics distribution for perishable commodities, consumers pay more attention to the pricing of commodity distribution services. Therefore, logistics distribution pricing strategy for perishable goods is a very important and crucial issue. On the one hand, if the logistics distribution is priced higher, it will lead to loss of customers; on the other hand, if the distribution price is too lower, the profit of the logistics

service provider will be impaired. Hence, the reasonable pricing strategy is a complex problem which needs to be taken into account carefully.

At present, there are many references focusing on the pricing strategies for perishable commodity distribution services. It can be divided into two aspects.

The first is developing an optimal inventory strategy for perishable goods. For example, Bai et al. [7] studied the inventory optimization problem under a carbon policy and constructed an inventory optimization model for perishable commodities under a carbon quota and trading policy. Ji et al. [8] studied a joint decision-making strategy for the inventory of perishable products in which the supplier's inventory was outsourced to a third party in the case of multiple suppliers and multiple vendors. Xiao et al. [9] used stochastic modeling and optimization methods to study the optimal unloading volume and optimal initial purchase volume decisions of the seller and its structural properties. Avinadav et al. [10] established an optimal pricing strategy, order quantity, and replenishment cycle model for perishable goods with price-dependent and time-varying requirements. Gutierrez-Alcoba [11] used an iterative algorithm to analyze and evaluate the implementation of inventory control for perishable goods. Kouki et al. [12] modeled the inventory system with zero inventory time as a Markov process, and proposed an iterative process for solving the problem of joint replenishment of perishable goods in the forward delivery period. Afshar-Nadjafi [13] studied the periodic inventory system of a perishable goods sales bulletin. Piramuthu et al. [14] considered that the demand for the inventory management of perishable goods directly depends on the allocation of shelf area of the item of interest and its instantaneous quality. Lee et al. [15] used a modified model to represent the system and consider the issuance policy to give a fixed order quantity, as well as joint ordering and issuance policy issues to study the inventory problem.

The references mentioned above aimed at inventory strategies for perishable goods, but during the process of perishable goods sales, logistics and distribution are more critical.

The second is research concentrating on pricing logistics services. For example, Li et al. [16] studied the pricing and efficiency of the secondary logistics service supply chain based on the principle of profit maximization under the condition of uncertain demand. Tian et al. [17] had the price form of split quotations and bundled quotations for online retail products and logistics services, considered the consumer's strategic behavior, and studied the applicable conditions and impact mechanism of the two quotations. Liu [18] used a method based on game theory to introduce the interaction between supply and demand in the pricing process into the pricing mechanism, thus forming a more reasonable third-party logistics service pricing mechanism. Li [19] established a related pricing model between different types of logistics enterprises based on the perspective of industrial integration, using the principles and methods of auction theory, combined with China's national conditions. Yu et al. [20] established two game theory models to explore the supply chain of fresh produce determined by the pricing and service level of a supplier and a retailer.

It can be seen from the above analysis that the second aspect is mainly for pricing logistics services, but perishable goods have certain special characteristics, including the distribution of perishable goods, hence more factors need to be considered. There are fewer references on the pricing strategy for logistics distribution services of perishable goods and factors related, such as delivery time and service satisfaction. In view of this, this paper adopts two pricing methods and simultaneously considers the distribution time and consumer service satisfaction. Consequently, four different pricing models are constructed, including a fixed pricing model, a fixed pricing model with time constraints, a dynamic pricing model, and a dynamic pricing model with time constraints. This paper compares and analyzes the characteristics of different pricing models and discusses the impact of them on the profit of logistics service providers, which provide a reference for the choice of pricing model for logistics service providers.

## 2. Problem Description and Symbol Representation

The research object of this paper is the logistics service providers of perishable goods. The logistics service providers can independently determine the logistics distribution pricing $P$. They also have to bear the main costs, including the unit cost $C$ of commodity distribution, the unit cost $h$ of commodity storage during the process of distribution, and the variable cost $W$ during the process of distribution. The distribution time is $T$, and the customer satisfaction of service is $l$.

The shorter the time spent by the logistics service provider during the distribution process, the lower the costs and the higher the consumer satisfaction, which will face more risk of deterioration [9]. Therefore, the delivery time is a variable. This paper studies the impact of different delivery times on different pricing models, and gives some advice to logistics service providers for pricing logistics distribution services. There are two notable characteristics during the distribution process of perishable goods:

(1) Periodicity, which means that the freshness of goods decreases with the increase of delivery time. Therefore, we define $V_t = V_0 e^{-\lambda t}$ as the value of perishable goods at time $t$ [21], here $V_0$ is the initial value of the perishable goods at $t = 0$, and $\lambda$ is the parameter of attenuation value.

(2) Distribution pricing and service satisfaction, which are related to the delivery quantity [22]. Consumers are more inclined to choose logistics service providers with higher cost performance. When the price is identical, consumers are more tend to choose providers with higher service satisfaction. The model of distribution pricing is defined as follows:

$$Q(t, P) = \frac{kl + V_t - P}{s} \tag{1}$$

Here, $s$ indicates the sensitivity of consumers to the cost performance of perishable goods, $k$ is the influence coefficient of service satisfaction on the quantity of goods. If and only if $P \leq kl + V_t$, the quantity of distribution is meaningful.

The main parameters (variables) used in the model are as follows: $P_i$ is the pricing decision variable under the fixed price and dynamic pricing strategies, $i = F$, $TF$, $D$, $TD$ represent the fixed pricing model, fixed pricing model with time constraints, dynamic pricing model and dynamic pricing model with time constraints, respectively. $\pi_i$ ($i = F$, $TF$, $D$, $TD$) indicates the average profit of the logistics service providers in the different models.

## 3. Model Construction of the Optimal Decision under Different Pricing Strategies

### 3.1. Optimal Fixed Pricing Model

The fixed pricing model means that the price is fixed and does not change even the decay of perishable goods and distribution time change [23]. Under the fixed pricing model, $P$ is a fixed value, and the average profit of the logistics service provider is defined as follows.

$$\pi_F = \frac{1}{T} \left[ \int_0^T (P - C)Q - hQdt - W \right] \tag{2}$$

where $(P - C)Q$ represents the total revenue of the logistics service provider considering the cost, $hQ$ represents the storage cost of deteriorating items in the distribution process, $W$ represents some additional expenses incurred in the delivery process, $\pi_F$ represents the average net profit of logistics providers in the fixed pricing model.

Substituting (1) into (2), the following formula can be obtained:

$$\pi_F = \frac{(kl + C + h)P - ckl - hkl - P^2}{s} - \frac{W}{T} + \frac{(P - C - h)}{sT} \int_0^T V_t dt$$

If $\frac{\partial \pi_F}{\partial P} = 0$, the distribution price makes the profit of logistics provider maximum. Correspondingly, it will be transferred as follows.

$$P_F{}^* = \frac{kl + h + C}{2} + \frac{\int_0^T V_t dt}{2T}$$

Substituting the value of the item at time *T*, formula will be obtained.

$$P_F{}^* = \frac{kl + h + C}{2} + \frac{V_0\left(1 - e^{-\lambda T}\right)}{2\lambda T} \tag{3}$$

However, under the fixed pricing model, logistics service providers do not take into account the delivery time of perishable goods, which easily reduces customer satisfaction because of delivery time delay and leads to a reduction in the number of customers, and ultimately reduces the overall profit of logistics service providers.

**Property 1.** *Under the fixed pricing model, the value of perishable goods decreases faster and the price for logistics distribution decreases.*

**Proof.** It can be known from (3) that $P_F{}^*$ is an expression about the decay parameter $\lambda$, and the first-order partial derivative of $\lambda$ is solved for (3), $\frac{\partial P_F{}^*}{\partial \lambda} = \frac{\left[(\lambda T+1)e^{-\lambda T}-1\right]V_0}{2T\lambda^2}$, with $0 < \lambda < 1$, $t > 0$. The order $f(\lambda) = (\lambda T + 1)e^{-\lambda T} - 1$, $\frac{\partial f(\lambda)}{\partial \lambda} = -\lambda T^2 e^{-\lambda T} < 0$ is established. Therefore, $f(\lambda)$ is a monotone decreasing function of $\lambda$, $f(0) = 0$, That is, $f(\lambda) < 0$ is established with $0 < \lambda < 1$. Also, $\frac{\partial P_F{}^*}{\partial \lambda} < 0$ is established with $0 < \lambda < 1$, so $P_F{}^*$ decreases with the increase of $\lambda$. $\square$

Property 1 reveals the relationship between distribution price and value decay rate of deteriorating commodities under a fixed pricing model. The value decay rate of perishable goods increases while the price of logistics distribution decreases. Therefore, the logistics service providers should select suitable perishable commodities with the decay index according to the scheduled delivery time for the maximum profit.

**Property 2.** *The higher the satisfaction of the consumers' logistics service, the higher the price of the logistics distribution.*

**Proof.** It can be shown from formula (3) that $P_F{}^*$ is an expression about consumer service satisfaction *l*, and the first-order partial derivative of *l* is adopted on formula (3); $\frac{\partial P_F{}^*}{\partial l} = \frac{k}{2} > 0$ is always true. In other words, $P_F{}^*$ is a monotonically increasing function on *l*. $\square$

**Property 3.** *The higher the cost of preservation of perishable goods, the higher the price of logistics and distribution.*

**Proof.** It can be seen from formula (3) that $P_F{}^*$ is an expression about the preservation cost *h* of a perishable commodity per unit time, and the first-order partial derivative of *h* is adopted on (3). The relation $\frac{\partial P_F{}^*}{\partial h} = \frac{1}{2} > 0$ is established. Namely, $P_F{}^*$ is a monotonically increasing function on *h*. $\square$

Under the fixed pricing model, it can be seen from the above properties that the pricing of perishable commodities is determined by many factors, such as consumer service satisfaction, the value attenuation parameters of perishable commodities, and the cost of perishable commodities. If the decay parameter of perishable goods is larger, the price of logistics distribution should be lower and the satisfaction degree of consumers should be higher. The price of logistics distribution should also be higher due to the higher storage cost of perishable goods. However, in real life, the fixed pricing model often makes consumers less satisfied with logistics services, which leads to a reduction in the

number of perishable goods that consumers decide to distribute, and the profits of logistics service providers will be reduced.

*3.2. Optimal Fixed Pricing Model with Time Constraints*

The fixed pricing model with time constraints is based on the fixed pricing model and considers the demand of consumers for distribution time, which meets the demand of consumers who have strict time constraints. Consumers are divided into ordinary consumers and strategy-based consumers. Strategy-based consumers refer to the distribution of perishable goods with specific requirements for time and price [24]. Strategic consumers often have certain requirements for delivery time. This paper considers a fixed pricing model with time constraints in order to realize the personalized service of perishable goods logistics distribution.

For the fixed pricing model with time constraints, the original planned delivery time is $T_1$, and the consumer expected time is $T_2$, let $T_0 = T_1 - T_2$. The variable cost of unit logistics is $C(l)$.

$$C(l) = k_0 T_0{}^2 \tag{4}$$

Here, $k_0$ is the coefficient of time-cost variation, $C(l)$ is a quadratic function of $T_0$. The greater the difference between the expected time of consumers and the original planned time, the greater the increase of the unit logistics distribution cost of perishable goods. Therefore, the linear relationship between $C(l)$ and $T_0$ is not used. The first derivative of $T_0$ for $C(l)$ is $\dot{C}(l) = 2k_0 T_0$. The results show that the rate of increase of distribution pricing is twice the original one. This is also in line with reality, namely the shorter the distribution time, the higher the increasing rate of distribution pricing. The relationship is more acceptable by consumers. The average profit of the logistics provider is defined as follows.

$$\pi_{TF} = \frac{1}{T_2}\left[\int_0^{T_2}(P - C - C(l))Q - hQdt - W\right] \tag{5}$$

Substituting (1) in to (5), the following can be derived:

$$\pi_{TF} = [P - C - C(l) - h]\frac{kl - P}{s} + \frac{P - C - C(l) - h}{sT_2}\int_0^{T_2}V_t dt - \frac{W}{T_2} \tag{6}$$

Substituting $V_t = V_0 e^{-\lambda t}$ and formula (4) into (6), if $\frac{\partial \pi_{TF}}{\partial P} = 0$ is satisfied, the distribution pricing makes the logistics service provider obtaining maximum profit, which can be shown in formula (7).

$$P_{TF}{}^* = \frac{kl + C + k_0 T_0{}^2 + h}{2} + \frac{V_0 - V_{T_2}}{2\lambda T_2} \tag{7}$$

**Property 4.** *The larger the difference between the expected time and the planned delivery time, the higher the distribution price.*

**Proof.** From (7), the larger the difference between the expected time and the planned delivery time, the greater the value of $T_0$. $P_{TF}{}^*$ is a quadratic function of $T_0$. For $P_{TF}{}^*$, the first order partial derivative of $T_0$ is obtained by $\frac{\partial P_{TF}{}^*}{\partial T_0} = k_0 T_0 > 0$ that is constant, here $P_{TF}{}^*$ is an incremental function of $T_0$. □

From (7), it can be seen that the pricing of the distribution service for perishable goods is determined by many factors, such as consumer service satisfaction, storage cost of perishable goods and distribution time difference. These factors have a positive effect on distribution pricing, which provides a reference for logistics service providers to formulate distribution pricing for perishable goods.

### 3.3. Optimal Dynamic Pricing Model

Dynamic pricing model differs from the fixed pricing model. It determines different distribution pricing according to the status of perishable goods at different times. Dynamic pricing is more akin to the competition-oriented pricing method [25], and adopt price differences and other means of operation to outcompete competitors and seize market share. The dynamic pricing model considers the decay of perishable goods during the process of distribution, the interests of consumers which are more reasonable, and adjusts the distribution pricing during the process of distribution. This will attract customers due to the more reasonable cost performance ratio, and then improves the overall interests of logistics service providers.

Under the dynamic pricing model, the average profit of logistics providers in a distribution cycle can be expressed as follows.

$$\pi_D = \frac{1}{T}\left[\int_0^T (P_D(t) - C)Q - hI_Q(t,T)dt - W\right] \tag{8}$$

Here, $I_Q(t,T)$ is the quantity of preservation of deteriorating commodities with the increase of delivery time. $I_Q(t,T) = \int_t^T Q(s,P)ds$. $I_Q(t,T)$ is related to the delivery price during delivery time.

$$\text{Suppose } f(t,P) = (P_D(t) - C)Q.$$

$$\text{Suppose } g(t,P) = hI_Q(t,T).$$

Then $\pi_D = \frac{1}{T}\left[\int_0^T f(t,P) - \int_t^T g(s,P)dsdt - W\right]$. Because the distribution pricing is based on the variable distribution time which is divided into $n$ $\Delta t$ ($\Delta t \to 0$). In the dynamic pricing model, the average profit can be defined as follows.

$$\pi_D = \lim_{\Delta s, \Delta t \to 0} \frac{1}{T}\left\{\sum_{t_i}^T \left[f(t_i, P_{t_i})\Delta t - \sum_{s_i=t_i}^T g(s_i, P_{s_i})\Delta s\right]\Delta t - W\right\}$$

For the first order partial derivative of the above formula for $P_{t_i}$, we substitute $\Delta t = \frac{T}{n}$ and $t_i = i\Delta t$, and we get $\frac{\partial \pi_D}{\partial P_{t_i}} = \frac{1}{T}\left\{\frac{\partial f(t_i, P_{t_i})}{\partial P_{t_i}} - \frac{\partial g(t_i, P_{t_i})}{\partial P_{t_i}}t_i\right\}$. Because $\frac{\partial f(t,P)}{\partial P} - \frac{\partial g(t,P)}{\partial P}t = 0$, the optimal dynamic pricing is expressed as follows.

$$P_D(t)^* = \frac{kl + C + V_t}{2} + \frac{ht(T-t)}{2} \tag{9}$$

From formula (9), it can be seen that the optimal dynamic pricing largely depends on the value of perishable goods at time $t$. The value of perishable goods decreases with delivery time, and the value of $P_D(t)^*$ decreases with the increase of delivery time.

**Property 5.** *If $\lambda^2 V_0 < h$, $\lambda T > 2$, with the delivery time, the optimal distribution pricing in the dynamic pricing model gradually increases and then decreases, and the extent of the reduction increases with the increase of the speed of commodity decay and the increase of storage cost.*

**Proof.** $\frac{\partial P_D(t)^*}{\partial t} = -\lambda V_t - ht + \frac{hT}{2}$, suppose $h(t) = -\lambda V_t - ht + \frac{hT}{2}$. For $h(t)$, the second derivative can be obtained, $\ddot{h}(t) = -\lambda^3 V_0 e^{-\lambda t} < 0$. $\dot{h}(t)$ is monotonically decreasing. $\dot{h}(t) = \lambda^2 V_0 e^{-\lambda t} - h$, $\lambda^2 V_0 < h$, $\dot{h}(0) = \lambda^2 V_0 - h < 0$, and then $\dot{h}(t) < 0$, so $h(t)$ is also monotonically decreasing. If $h(0) = -\lambda V_0 + \frac{Th}{2} > -\lambda V_0 + \frac{\lambda^2 V_0 T}{2} = \lambda V_0\left(\frac{\lambda T}{2} - 1\right) > 0$, so $\frac{\partial P_D(t)^*}{\partial t}$ is greater than 0 at first and then less than 0 with the increase of $t$. The optimal distribution pricing gradually rises and then decreases. With the increase of $\lambda$ and $h$, the distribution price declines further. □

*3.4. Optimal Dynamic Pricing Model with Constrained Time*

The dynamic pricing model with time constraints is based on the dynamic pricing model, which takes into account the demand of consumers for distribution time and meets consumers' strict requirements.

For the fixed pricing model with time constraints, the original planned delivery time is $t_1$, and the consumer expected time is $t_2$. Suppose $t_0 = t_1 - t_2$. The variable cost of unit logistics is $c(l)$.

$$c(l) = k_1 t_0^2 \tag{10}$$

where $k_1$ is the time cost variation coefficient. In the dynamic pricing model with time constraints, the average profit of the logistics service provider in the distribution cycle can be expressed as follows.

$$\pi_{TD} = \frac{1}{t_2}\left[\int_0^{t_2}(P_{TD}(t) - C - c(l)Q - hQ_Q(t,T)dt - W)\right] \tag{11}$$

Here, $Q_Q(t, T)$ is the preserved quantity of perishable goods at time $t$. At this point, the following definition can be given.

$$Q_Q(t,T) = \int_t^T Q(s,P)ds \tag{12}$$

Suppose $f(t,P) = (P_D(t) - C - c(l))Q$

Suppose $g(t,P) = hQ_Q(t,T)$

Then

$$\pi_{TD} = \frac{1}{t_2}\left[\int_0^{t_2} f(t,P) - \int_0^{t_2} g(s,P)dsdt - W\right]$$

Since price of dynamic pricing model with the time-constrained is variable, the delivery time is divided into $n$ $\Delta t$ ($\Delta t \to 0$). The average profit in the dynamic pricing model can be expressed as follows.

$$\pi_{TD} = \lim_{\Delta s, \Delta t \to 0} \frac{1}{t_2}\left\{\sum_{t_i}^{t_2}\left[f(t_i, P_{t_i})\Delta t - \sum_{s_i=t_i}^{t_2} g(s_i, P_{s_i})\Delta s\right]\Delta t - W\right\}$$

$$= \lim_{\Delta t \to 0} \frac{1}{t_2}\{[f(t_1, P_{t_1}) \quad + f(t_2, P_{t_2}) + f(t_3, P_{t_3}) + \cdots + f(t_n, P_{t_n})]\Delta t$$

$$- [g(t_1, P_{t_1}) + g(t_2, P_{t_2}) + g(t_3, P_{t_3}) + \cdots + g(t_n, P_{t_n})]\Delta t^2$$

$$- [g(t_2, P_{t_2}) + g(t_3, P_{t_3}) + \cdots + g(t_n, P_{t_n})]\Delta t^2 - g(t_n, P_{t_n})\Delta t^2 - W\}$$

Solving the first-order partial derivative of $P_{t_i}$ for the equation mentioned above, and substituting $\Delta t = \frac{T}{n}$ and $t_i = i\Delta t$, we can obtain:

$\frac{\partial \pi_D}{\partial P_{t_i}} = \frac{1}{t_2}\left\{\frac{\partial f(t_i, P_{t_i})}{\partial P_{t_i}}\Delta t - i\frac{\partial g(t_i, P_{t_i})}{\partial P_{t_i}}\Delta t^2\right\} = \frac{1}{t_2}\left\{\frac{\partial f(t_i, P_{t_i})}{\partial P_{t_i}} - \frac{\partial g(t_i, P_{t_i})}{\partial P_{t_i}}t_i\right\}$. Because $\frac{\partial f(t,P)}{\partial P} - \frac{\partial g(t,P)}{\partial P}t = 0$, the optimal dynamic pricing is expressed as follows.

$$P_{TD}(t)^* = \frac{kl + C + V_t + k_1 t_0^2}{2} + \frac{htt_0}{2} \tag{13}$$

**Property 6.** *The shorter the expected delivery time, the higher the distribution price.*

**Proof.** The shorter the time of consumer expects, the larger the value of $t_0$. Formula (13) shows that $P_{TD}(t)^*$ is a function of $t_0$, and the first-order partial derivative can be obtained for $t_0$, $\frac{\partial P_{TD}^*}{\partial t_0} = k_1 t_0 + \frac{ht}{2} > 0$, which is always true, namely, $P_{TD}(t)^*$ is an increasing function on $t_0$. □

## 4. Comparison Analysis of Four Pricing Models

*4.1. Comparison of Optimal Delivery Pricing*

Because the distribution pricing model with time constraints adds time constraints to the original model, this paper only compares the maximum distribution pricing between the fixed pricing model and the dynamic pricing model.

**Proposition 1.** *Under the same distribution time, if $0 < t < \bar{t}$, then the distribution pricing under the dynamic pricing model is higher than that under the fixed pricing model. If $\bar{t} < t < T$, the distribution price under the dynamic pricing model is lower than the fixed price model. Here, the parameter $\bar{t}$ satisfies $V_{\bar{t}} + h\bar{t}(T - \bar{t}) = h + \frac{(V_0 - V_T)}{\lambda T}$, and $T > \bar{t} > \frac{T}{2}$.*

**Proof.** We compare formulas (3) and (9), and consider the description of the first increase and decrease of $P_D(t)^*$ in property 5. Firstly, compare $P_D(0)^*$ with $P_F^*$. $P_D(0)^* = \frac{kl+C+V_0}{2}$, wihle $t = 0$, we get $P_F^* = \frac{kl+h+C}{2} + \frac{V_0\left(1-e^{-\lambda T}\right)}{2\lambda T}$. If $h > V_0 + \frac{V_T-V_0}{\lambda T}$, $P_F^* > P_D(0)^*$, otherwise, $P_F^* < P_D(0)^*$. Let $y(t) = \frac{kl+C+V_0}{2} - \frac{kl+h+C}{2} + \frac{V_0\left(1-e^{-\lambda T}\right)}{2\lambda T}$, which can be transferred to $y(t) = \frac{V_0-h}{2} - \frac{V_0-V_T}{2T\lambda} = \frac{\lambda T V_0 - h\lambda T - V_0 + V_T}{2\lambda T} = \frac{(\lambda T-1)V_0 - h\lambda T + V_T}{2\lambda T} > \frac{V_0 + V_T - h\lambda T}{2\lambda T} > 0$, in other words $P_F^* < P_D(0)^*$. Because $P_D(t)^*$ monotonically increases if $0 < t < \frac{T}{2}$ and $P_F^*$ remains unchanged, $P_F^* < P_D(t)^*$ holds if $0 < t < \frac{T}{2}$. If $t = T$, $P_F^* = \frac{kl+h+C}{2} + \frac{V_0\left(1-e^{-\lambda T}\right)}{2\lambda T}$, $P_D(T)^* = \frac{kl+C+V_T}{2}$, let $u(t) = \frac{kl+h+C}{2} + \frac{V_0\left(1-e^{-\lambda T}\right)}{2\lambda T} - \frac{kl+C+V_T}{2}$, then $u(t) = \frac{h-V_T}{2} + \frac{V_0\left(1-e^{-\lambda T}\right)}{2\lambda T} = \frac{h\lambda T - \lambda T V_T + V_0 - V_T}{2\lambda T} = \frac{V_0 - (\lambda T+1)V_T + h\lambda T}{2\lambda T} > 0$, if $t = \frac{T}{2}$, $P_F^* < P_D\left(\frac{T}{2}\right)^*$. If $t = T$, $P_F^* > P_D(T)^*$, so there is $\frac{T}{2} < \bar{t} < T$. If $0 < t < \bar{t}$, the distribution price under the dynamic pricing model is higher than that under the fixed pricing model. On the contrary, If $\bar{t} < t < T$, the distribution price under the dynamic pricing model is lower than the fixed price model. □

*4.2. Comparison of Applied Range*

The factors affecting the distribution pricing of perishable goods under the fixed pricing model are consumer service satisfaction and the decay index of commodities. The characteristic of this pricing model is constant distribution price during the distribution time. A fixed pricing model will lead to longer delivery times and consumers will give a lower degree of service satisfaction, which will lead to a reduction in the number of consumers, and then affects the overall profit of logistics service providers. A fixed pricing model with time constraints has largely reduced the decline of consumer service satisfaction caused by long delivery times. From Section 3.1, it can be seen that the commodity decay index increases and the logistics distribution pricing decreases under the fixed pricing model. Therefore, it is very important for the logistics service provider to choose the appropriate commodity decay index according to the distribution time in this mode.

The factors that influence the price of the dynamic pricing model are consumer service satisfaction, storage cost of perishable goods, and their decay index. A dynamic pricing model distributes perishable goods with dynamic price, which avoids the reduction of distribution quantity caused by a decrease in customer service satisfaction. Therefore, this pricing model is more suitable under these circumstances. Under the dynamic pricing model, distribution price decreases with the increase of distribution time, decay index and storage cost of perishable goods. The larger the increase of decay index and storage cost of perishable goods, the more the distribution pricing decreases.

## 5. Experiment Analysis

This section validates the proposed four strategies through numerical comparisons. Suppose that the initial value of perishable goods is $V_0 = 20$ RMB, with distribution time $T = 20$ h, decay index of perishable goods $\lambda = 0.01$, the distribution cost per unit of perishable goods $C = 4$ RMB, demand

parameters of perishable goods $s = 2$, $kl = 10$, and the storage cost per unit time of perishable goods $h = 0.05$ RMB, and $W = 50$ RMB.

## 5.1. Results of Coparing Different Models

Suppose that for the fixed pricing models with time constraints and the dynamic pricing models with time constraints, consumers require a three-hour reduction in delivery time and a time cost variation factor of 0.01. Then, the optimal price and average profit in the four models are shown in Table 1.

**Table 1.** Optimal price and average profit in four models.

| No | Four Models | Optimal Price (RMB) | Average Profit (RMB) |
|---|---|---|---|
| 1 | Fixed pricing model | $P_F^* = 16.09$ | 69.96 |
| 2 | Fixed pricing model with time constraints | $P_{TF}^* = 16.27$ | 70.58 |
| 3 | Dynamic pricing model | $P_D(0)^* = 17.00$ | 67.19 |
| 4 | Dynamic pricing model with time constraints | $P_{TD}(0)^* = 17.0004$ | 70.77 |

As shown in Table 1, the optimal price and average profit of the pricing models with time constraints are higher than those of the pricing model without time constraints. At $t = 0$, $P_F^* < P_D(0)^*$. The average profit of the dynamic pricing model with time constraint is higher than other three pricing models.

## 5.2. Results of the Fixed Pricing Model

Under the fixed pricing model, the decay indexes of perishable commodity are 0.05, 0.10, 0.15, 0.20, respectively, and the distribution time $T$ ranges from 0 to 20 in steps of 0.25. The corresponding distribution pricing is shown in Figure 1. The values for $T$ are 2, 4, 6, 8, 10, 12, 14, 16, 18 and 20 corresponding to Table 2.

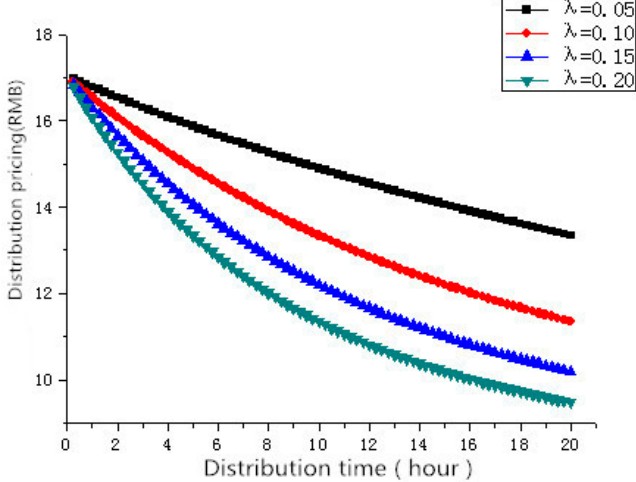

**Figure 1.** Relationship between distribution pricing and $(\lambda, T)$ in the fixed pricing model.

**Table 2.** Relationship between distribution pricing and $(\lambda, T)$ in the fixed pricing model.

| Distribution Time / Value Decay Index | 2 | 4 | 6 | 8 | 10 | 12 | 14 | 16 | 18 | 20 |
|---|---|---|---|---|---|---|---|---|---|---|
| $\lambda = 0.05$ | 16.54 | 16.09 | 15.66 | 15.27 | 14.89 | 14.54 | 14.22 | 13.91 | 13.62 | 13.35 |
| $\lambda = 0.10$ | 16.09 | 15.27 | 14.54 | 13.91 | 13.35 | 12.85 | 12.41 | 12.01 | 11.66 | 11.35 |
| $\lambda = 0.15$ | 15.66 | 14.54 | 13.62 | 12.85 | 12.2 | 11.66 | 11.2 | 10.81 | 10.48 | 10.19 |
| $\lambda = 0.20$ | 15.27 | 13.91 | 12.85 | 12.01 | 11.35 | 10.81 | 10.38 | 10.02 | 9.73 | 9.48 |

As shown in Figure 1 and Table 2, the distribution pricing decreases with the increase of the decay index, which is similar to the conclusion of property 1. The longer the delivery time, the lower the distribution price is. When the logistics service provider chooses the fixed pricing model, it should pay more attention to the selection of perishable goods with smaller decay index for distribution. For perishable goods with a larger decay index, it should reduce the distribution time as far as possible to reduce the risk caused by the decay of such goods in the distribution process.

According to property 2, the service satisfaction of consumers is equivalent to the price of logistics distribution. Here, values of $l$ are 0.2, 0.4, 0.6 and 0.8 and the distribution time $T$ is in 0.25 increments from 0 to 20. The corresponding distribution time is 2, 4, 6, 8, 10, 12, 14, 16, 18, and 20. The results are shown in Figure 2 and Table 3.

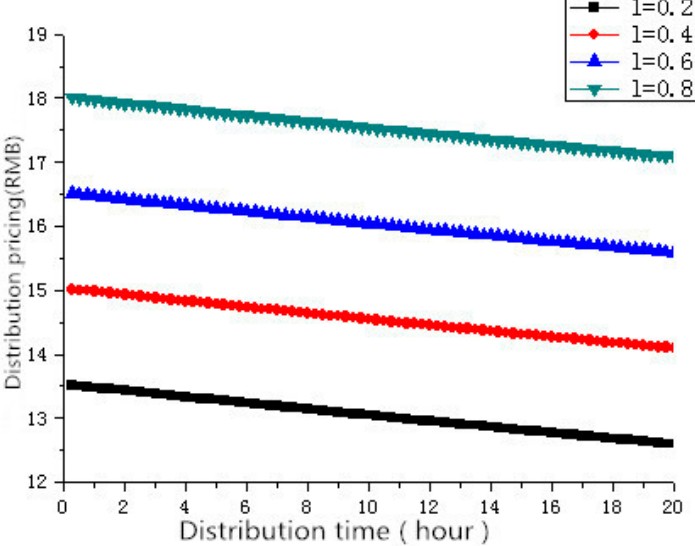

**Figure 2.** Relationship between distribution pricing and ($l$, $T$) in the fixed pricing model.

**Table 3.** Relationship between distribution pricing and ($l$, $T$) in the fixed pricing model.

| Distribution Time<br>Service Satisfaction | 2 | 4 | 6 | 8 | 10 | 12 | 14 | 16 | 18 | 20 |
|---|---|---|---|---|---|---|---|---|---|---|
| $l = 0.2$ | 13.43 | 13.33 | 13.23 | 13.14 | 13.04 | 12.95 | 12.86 | 12.77 | 12.68 | 12.59 |
| $l = 0.4$ | 14.93 | 14.83 | 14.73 | 14.64 | 14.54 | 14.45 | 14.36 | 14.27 | 14.18 | 14.09 |
| $l = 0.6$ | 16.43 | 16.33 | 16.23 | 16.14 | 16.04 | 15.95 | 15.86 | 15.77 | 15.68 | 15.59 |
| $l = 0.8$ | 17.93 | 17.83 | 17.73 | 17.64 | 17.54 | 17.45 | 17.36 | 17.27 | 17.18 | 17.09 |

From Figures 2 and 3, we can see that the higher the satisfaction degree of consumer service, the higher the distribution pricing of perishable goods, which is same to the conclusion of property 2. Therefore, the logistics service provider should accomplish the distribution within the prescribed time as far as possible, and ensure the freshness of perishable goods in the distribution process. In this way, consumer satisfaction will be enhanced, which would then, in turn, improve the distribution pricing of perishable goods.

### 5.3. Results of the Fixed Pricing Model with Time Constraints

Under the fixed pricing model with time constraints, suppose that the original plan delivery time $T$ is 20 h, and the difference between $T_0$ and 10 is 0.25, the corresponding distribution pricing is shown in Figure 3.

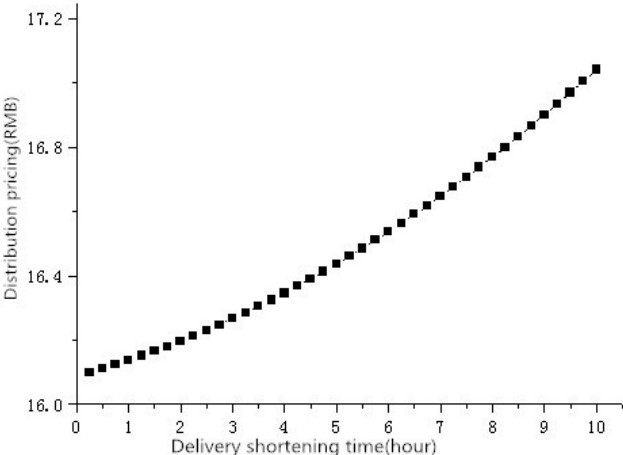

**Figure 3.** The relationship between distribution pricing and $T_0$ under fixed pricing with time constraints.

### 5.4. Results of the Dynamic Pricing Model

Under the dynamic pricing model, the main parameters of the distribution pricing are consumer satisfaction, the attenuation coefficient of perishable goods, storage cost and distribution time. This section mainly demonstrates the impact of distribution time, attenuation coefficient and storage cost of perishable commodities on distribution pricing.

In this paper, the expected delivery time $T$ is 20 h, the initial value of the goods $V_0$ is 20 RMB, values of $\lambda$ are 0.1, 0.2, 0.3, and values of $h$ are 0.5 and 0.7. The corresponding distribution pricing is shown in Figure 4 and Table 4.

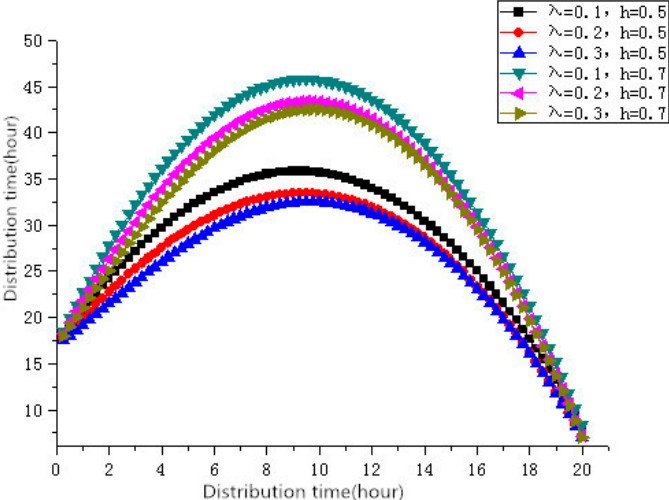

**Figure 4.** Relationship between $(\lambda, h)$ and distribution pricing under dynamic pricing model.

**Table 4.** Relationship between distribution pricing and $(\lambda, h)$ under dynamic pricing.

| Time (λ, h) | 2 | 4 | 6 | 8 | 10 | 12 | 14 | 16 | 18 | 20 |
|---|---|---|---|---|---|---|---|---|---|---|
| $\lambda = 0.1, h = 0.5$ | 24.19 | 29.70 | 33.49 | 35.49 | 35.68 | 34.01 | 30.47 | 25.02 | 17.65 | 8.35 |
| $\lambda = 0.2, h = 0.5$ | 22.70 | 27.49 | 31.01 | 33.02 | 33.35 | 31.91 | 28.61 | 23.41 | 16.27 | 7.18 |
| $\lambda = 0.3, h = 0.5$ | 21.49 | 26.01 | 29.65 | 31.91 | 32.50 | 31.27 | 28.15 | 23.08 | 16.05 | 7.02 |
| $\lambda = 0.1, h = 0.7$ | 27.79 | 36.10 | 41.89 | 45.09 | 45.68 | 43.61 | 38.87 | 31.42 | 21.25 | 8.35 |
| $\lambda = 0.2, h = 0.7$ | 26.30 | 33.89 | 39.41 | 42.62 | 43.35 | 41.51 | 37.01 | 29.81 | 19.87 | 7.18 |
| $\lambda = 0.3, h = 0.5$ | 25.09 | 32.41 | 38.05 | 41.51 | 42.50 | 40.87 | 36.55 | 29.48 | 19.65 | 7.02 |

As shown in Figure 4 and Table 4, the dynamic distribution price increases first and then decreases with the delivery time. By comparing the curve $\lambda = 0.1$, $h = 0.5$ and the curve $\lambda = 0.2$, $h = 0.7$, we find that the distribution pricing of the curve $\lambda = 0.2$, $h = 0.7$ decreases faster, in other words, the optimal distribution pricing increases gradually and then decreases with the time of distribution, and the extent of the reduction increase with the decay rate of the value of perishable goods and the cost of preservation. This is consistent with the conclusion of property 5.

### 5.5. Results of the Dynamic Pricing Model with Time Constraint

Under the dynamic pricing model with time constraints, suppose that the original planned delivery time $T$ is 20 h, and $t_0$ ranges from 0 to 10 in increments of 0.25. The corresponding delivery pricing is shown in Figure 5.

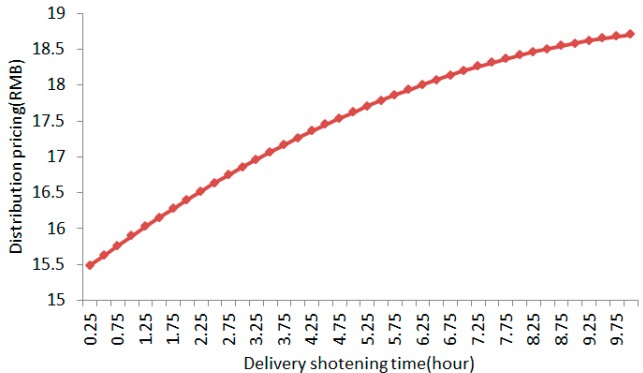

**Figure 5.** Relationship between distribution pricing and shortening time under time constrained dynamic pricing model.

Figure 5 shows that the larger the difference between the expected time and the original distribution time, the higher the distribution pricing, which is consistent with the conclusion of property 6.

## 6. Conclusions

This paper has presented approaches for handling the decay in perishable goods distribution as well as the sensitivity characteristics of consumers to delivery time and service satisfaction, using four different pricing models that are the fixed pricing model and dynamic pricing model with and without time constraints, respectively. Then, the profit model of the logistics service provider is constructed and the corresponding optimal pricing and average profit are obtained. The influences of the decay index, delivery time, and consumer service satisfaction on distribution pricing are analyzed. The following conclusions are obtained.

(1) The relationship between the optimal price and the decay index under four pricing models are as follows: Under the fixed pricing model, the larger the decay index, the lower the distribution pricing of perishable commodities; under the fixed pricing model with time constraints, the larger the decay index, the lower the distribution pricing; and the dynamic pricing model has the lowest distribution pricing. Optimal distribution price first rises and then decreases, at the same time, the greater the decay index, the greater the extent of decline when the optimal distribution pricing drops. Under the dynamic pricing model with time constraints, the relationship between the decay index and the optimal pricing is same to that under the dynamic pricing model.

(2) The relationship among optimal price, customer service satisfaction, and storage cost of perishable goods under the four pricing models are given as follows: Under the fixed pricing model and fixed pricing model with time constraints, the optimal price increases with the increase of customer service satisfaction and storage cost; under the dynamic pricing model and dynamic pricing model with time constraints, the optimal price increases first and then decreases. Within the increasing range

of distribution price, the distribution pricing increases with the increase of customer service satisfaction and storage cost. Within the decreasing range of distribution price, the distribution price increases with the increase of customer service satisfaction and storage cost. The greater the increase in consumer service satisfaction and storage costs, the greater the reduction in distribution price.

(3) Comparing the fixed pricing model with the dynamic pricing model, we found that if $\bar{t}$ satisfies $V_{\bar{t}} + h\bar{t}(T - \bar{t}) = h + \frac{(V_0 - V_T)}{\lambda T}$, the distribution price under the dynamic pricing model is higher than that under the fixed pricing model for $0 < t < \bar{t}$. For $\bar{t} < t < T$, the distribution price under the dynamic pricing model is lower than that under the fixed price model.

(4) Comparing the four pricing models, we found that the average profit of the logistics service provider under the fixed pricing model with time constraints is higher than that under the fixed pricing model. The average profit of the logistics service provider under the dynamic pricing model with time constraints is higher than that under the dynamic pricing model alone. The average profit of the logistics service provider under the dynamic pricing model with time constraints is higher than that under other three models.

In addition to these conclusions, the following suggestions are given for logistics providers:

(1) Consumers are more inclined to choose the dynamic pricing model either with or without time constraints. For two dynamic models mentioned above, consumer service satisfaction is more important to the quantity of distribution, and it indirectly affects the overall profit. Therefore, for these two models, logistics service providers should pay more attention to the response of consumers, or take more active measures to improve their own consumer service satisfaction.

(2) Logistics service providers can choose mass distribution of perishable commodities under the same storage conditions, which will reduce the storage cost of perishable commodities per unit in the distribution process, and then increases their own profits.

Pricing models of logistics distribution services for perishable commodities don't consider the influence of season, weather, distribution mode of transportation, deliver region, etc. and relationships among them, which are also crucial and should be studied in future.

**Author Contributions:** conceptualization, T.L. and Y.C.; methodology, T.L.; validation, T.L.; formal analysis, T.L.; data curation, T.L.; writing—original draft preparation, T.L.; writing—review and editing, T.L.; visualization, T.L.; supervision, Y.C.; project administration, T.L.; funding acquisition, T.L.

**Funding:** This research was funded by the National Science and Technology Major Project of China, grant number 2017YFC1404602.

**Conflicts of Interest:** The authors declare no conflict of interest.

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
