# Peer review of "Pricing Strategies of Logistics Distribution Services for Perishable Commodities"

_algorithms, doi:10.3390/a11110186_

Reviewer 1 Report

Dear Authors,

I really enjoyed reading the manuscript. Although, this study is quite common and to my opinion contribution to extant literature is minimum. However, despite of limited contribution to theory, I still believe that this article has enough merit to contribute to shape decision making process of the managers engaged in design and management of supply chain for perishape goods. I understand that authors have jumped directly to the problem and research questions. However, I invite authors to build an interesting story using some classical papers focussing on design of supply chains like:

Fisher, M., Hammond, J., Obermeyer, W., & Raman, A. (1997). Configuring a supply chain to reduce the cost of demand uncertainty. Production and operations management6(3), 211-225.

Fisher, M.L. (1997). What is the right supply chain for your product?. Harvard Business Review, March-April (please check the page number using SCOPUS database to avoid confusion).

Lee, H. L. (2002). Aligning supply chain strategies with product uncertainties. California management review44(3), 105-119.

Ali, S. S., Madaan, J., Chan, F. T., & Kannan, S. (2013). Inventory management of perishable products: a time decay linked logistic approach. International Journal of Production Research51(13), 3864-3879.

Dubey, R., Gunasekaran, A., & Childe, S. J. (2015). The design of a responsive sustainable supply chain network under uncertainty. The International Journal of Advanced Manufacturing Technology80(1-4), 427-445.

I hope these papers will surely help to build an interesting stories that how nature of the product and  environmental uncertainties, influence the design of supply chains. Overall, I am satisfied with the model and the analysis which authors have performed. The presentation needs to be upgraded to enhance the reading experience of the readers.

Author Response

Point 1: I invite authors to build an interesting story using some classical papers focussing on design of supply chains.

Response 1:  We have revised the paper and rewrite the introduction. We only focus on the issue of pricing logistics distribution services for perishable goods, and other issues will not be covered in this paper.

Point 2: The presentation needs to be upgraded to enhance the reading experience of the readers.

Response 2:  We have asked a special translation company to help us modify the language.

Reviewer 2 Report

I think that it is an interesting paper that deals with different models of pricing for perishable goods leading to a profit model of the logistics service provider.  Therefore an optimal pricing and average profit are identified. The evidence through experimental analysis is also provided. Also the paper is providing a few recommendations for logistics services provider leading at accurate / optimal pricing models for perishable goods. However I should recommend to include the limitations of the developed models and additional details on future directions. 

A detailed background of the existing research and modelling approaches leading to this paper's contributions and originality will enhance the value of the paper.

Generally it is well written but I have identified some inconsistency and typos mistakes and hence a careful checking of academic writing is needed.  

Author Response

Point 1: I should recommend to include the limitations of the developed models and additional details on future directions. 

Response 1:  We have added the limitations and possible trend of the research in  in the last paragraph of the conclusion.

Point 2: A detailed background of the existing research and modelling approaches leading to this paper's contributions and originality will enhance the value of the paper.

Response 1:  We have rewritten the introduction.

Point 3: I have identified some inconsistency and typos mistakes and hence a careful checking of academic writing is needed. 

Response 3:  We have asked a special translation company to revise the language and already revised the paper again.

Round  2

Reviewer 1 Report

Thanks for addressing my queries.